# An examination of the prevalence of metabolic syndrome in older adults in Ireland: Findings from The Irish Longitudinal Study on Ageing (TILDA)

**Kevin McCarthy**[1,2]*, **Eamon Laird**[1], **Aisling M. O'Halloran**[1], **Padraic Fallon**[1], **Deirdre O'Connor**[1], **Román Romero Ortuño**[1,2], **Rose Anne Kenny**[1,2]

**1** School of Medicine, Trinity College Dublin, Dublin, Ireland, **2** Mercer's Institute for Successful Ageing, St James's Hospital, Dublin, Ireland

* mccartk8@tcd.ie

**Data Availability Statement:** TILDA data is publicly available, at no monetary cost, via the Irish Social Science Data Archive (www.ucd.ie/issda). The

## Abstract

Metabolic syndrome (MetS) consists of the cluster of central obesity, insulin resistance, hypertension and atherogenic dyslipidaemia. It is a risk factor for cardiovascular disease, diabetes, and mortality. The prevalence of MetS has not been described in older adults from a population-representative sample in a European country before. This study aimed to determine the prevalence of MetS in older adults in Ireland and examine the association between MetS and socio-demographic, health, and lifestyle factors. This study used data from a population aged ≥50 years from waves 1 and 3 of the Irish Longitudinal Study on Ageing. The prevalence of MetS using the National Cholesterol Education Program Third Adult Treatment Panel (ATPIII) and the International Diabetes Foundation (IDF) criteria were determined. Weighted logistic regression examined the association between MetS and age, sex, education, and physical activity. MetS status was determined at both waves with transitions examined. 5340 participants had complete data for MetS criteria at wave 1. 33% had MetS according to the ATPIII criteria (32.5%; 95% CI: 31.1, 34.0), with 39% according to the IDF criteria (39.3%; 95% CI: 37.8, 40.8). MetS was more prevalent with advancing age, among males, those with lower educational attainment and lower physical activity. 3609 participants had complete data for both waves– 25% of those with MetS at wave 1 did not have MetS at wave 3 but the overall number of participants with MetS increased by 19.8% (ATPIII) and 14.7% (IDF). MetS is highly prevalent in older adults in Ireland. 40% of the 1.2 million population aged ≥50 years in Ireland meet either the ATPIII or IDF criteria. Increasing age, male sex, lower educational attainment, and lower physical activity were all associated with an increased likelihood of MetS.

## Introduction

Metabolic syndrome (MetS) is described as the cluster of inter-related cardiovascular risk factors of metabolic origin, occurring together more often than by chance alone, specifically the

publicly accessible dataset files are hosted by the
Irish Social Science Data Archive based in
University College Dublin, and the Interuniversity
Consortium for Political and Social Research
(ICPSR) based in the University of Michigan.
Researchers wishing to access the data must
complete a request form, available on either the
ISSDA (http://www.ucd.ie/issda/data/tilda/) or
ICPSR website (http://www.icpsr.umich.edu/
icpsrweb/ICPSR/studies/34315).

**Funding:** TILDA is funded by Atlantic
Philanthropies, the Irish Department of Health, Irish
Life plc. and the Health Research Board. The
funders did not have any involvement in study
design; in the collection, analysis, and
interpretation of data; in the writing of the report;
and in the decision to submit the paper for
publication.

**Competing interests:** The authors have declared
that no competing interests exist.

presence of a combination of 3 or more of: central obesity, insulin resistance (IR), hypertension, elevated triglycerides (TG), and/or reduced high-density lipoprotein (HDL) [1].

MetS is a prothrombotic, proinflammatory state and is recognised risk factor for type 2 diabetes mellitus (diabetes), cardiovascular disease (CVD), non-alcoholic fatty liver disease, and several cancers. Meta-analyses have shown MetS to be associated with a 1.58-fold increased risk for all-cause mortality [2–4].

The original rationale for diagnosing MetS was to identify those who are at high risk of developing CVD and diabetes that may otherwise not be identified, given those with MetS have additional cardiovascular risk over and above the individual risk factors [5].

MetS is a cluster of different conditions, rather than a single disease, and consequently there have been many names given to it and numerous criteria used to define it [6], with estimates of the prevalence of MetS differing depending on the definition used and the age, sex and race of the population examined and when studies were reported. Prevalence estimates have ranged from <5% in young adults and children to >80% in men with diabetes [7], while age-adjusted prevalence in the United States was found to have increased by 12% between studies approximately 10 years apart, with a far more marked increase among women (23.5%) than men (2.2%) [8].

In this study we examined two widely used MetS diagnostic criteria: the National Cholesterol Education Program Third Adult Treatment Panel (ATPIII) [9] and the International Diabetes Federation (IDF) [10].

Many estimates of prevalence of MetS in Europe have used samples in middle-age or older adults but with upper age-limits. None have examined prevalence specifically among older adults from age ≥50 years, without an upper age-limit, using a population-representative sample. The prevalence of MetS in Ireland has not been comprehensively described in a large population-representative sample before. It has been examined in Ireland previously, in a screening population of 1716 adults aged 32 to 78 years, where the prevalence of MetS was 13.2% and 21.4% for ATPIII and IDF criteria respectively [11]. It has also been investigated in a sample of 1018 adults aged between 50 and 69 years using the World Health Organisation (WHO) criteria with a prevalence of 21% [12], and also in subpopulations including those with diabetes, schizophrenia and Irish travellers [13–15]. Thus, very little is known about the prevalence of MetS among older adults in Ireland using ATPIII and IDF criteria. In this study we aimed to characterise and determine the national prevalence of MetS in older adults in Ireland, using both the ATPIII and IDF criteria, using data from the first wave of The Irish Longitudinal Study on Ageing (TILDA), a prospective study of the health, social and economic circumstances, designed to be representative of community-dwelling adults aged ≥50 years in Ireland. We also aimed to examine how those with and without MetS progressed longitudinally at a 4-year follow-up.

## Materials and methods

### Sample

This observational study is based on data from the first wave of TILDA (n = 8173), collected between October 2009 and February 2011, with the data collection process being described in detail elsewhere [16, 17]. In short, as part of the study participants completed a Computer Assisted Personal Interview (CAPI), conducted in participants' homes by trained interviewers, and a health assessment (HA), including blood draws, carried out by research nurses in one of two centres. Those <50 years at wave 1, and those who did not complete the HA were excluded. For the longitudinal analyses, data from wave 3 of TILDA were used. This was

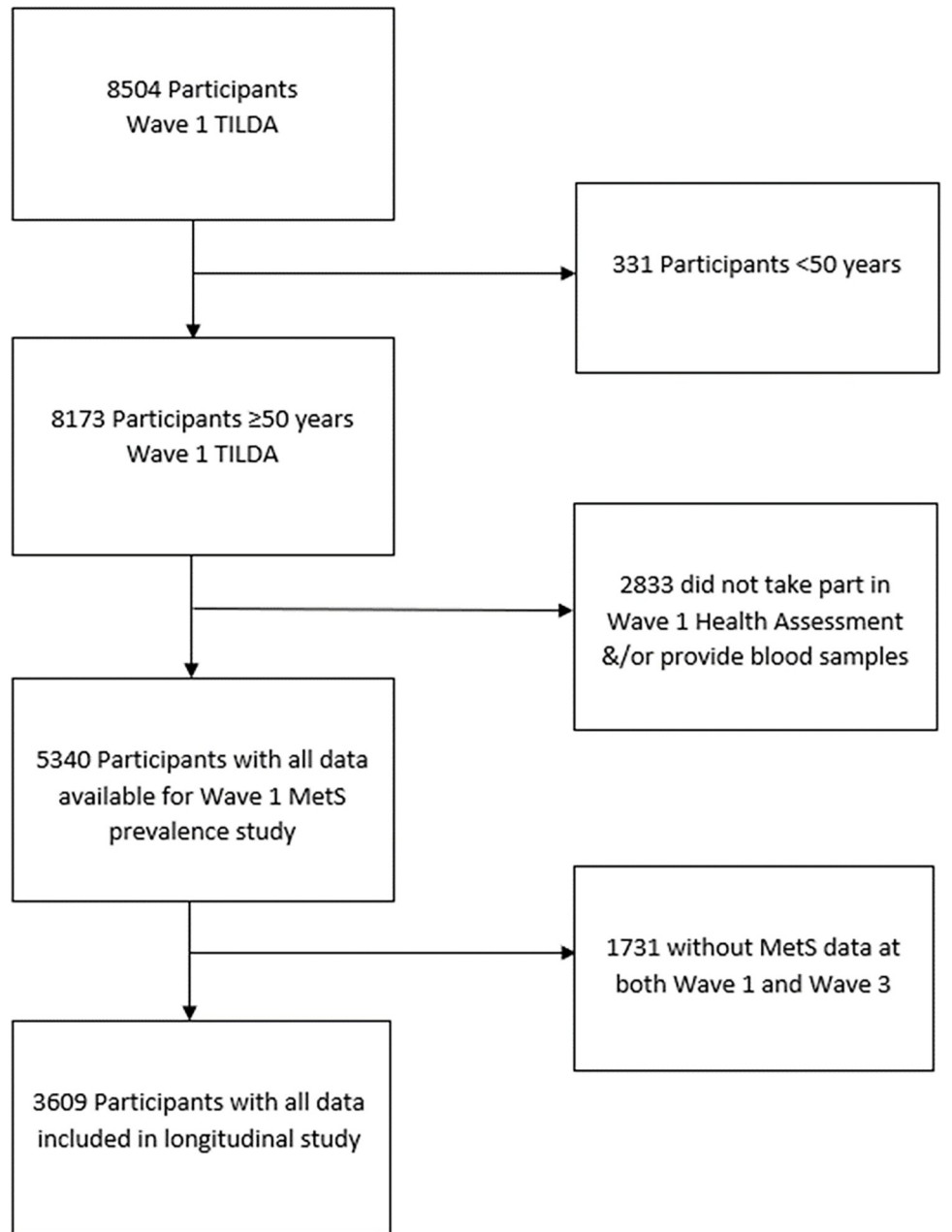

**Fig 1. Flowchart of study participants.**

collected between March 2014 and October 2015. Those who were ≥50 years at wave 1 and had all relevant CAPI and HA data at both waves were included in these analyses (Fig 1).

## Metabolic syndrome criteria

Two sets of criteria were used to calculate prevalence: ATPIII and IDF criteria. Components of MetS were measured using objective data from the HA; specifically, waist circumference (WC), TG, HDL, systolic (SBP) and diastolic blood pressure (DBP) and glycated haemoglobin (HbA1c), as well as self-reported doctor-diagnosed medical conditions and regular medications (CAPI).

**Table 1. ATPIII and IDF criteria for diagnosis of metabolic syndrome.**

| | ATPIII | IDF |
|---|---|---|
| **Central Obesity** | WC >102 cm (male) or >88 cm (female) | BMI >30 kg/m$^2$ or WC ≥94 cm (male) or ≥80 cm (female) |
| **Insulin Resistance[a]** | Raised fasting glucose (≥5.6 mmol/L) | Raised fasting glucose (≥5.6 mmol/L) |
| **Blood Pressure** | SBP≥130 mmHg or DBP≥85 mmHg, or treatment | SBP≥130 mmHg or DBP≥85 mmHg, or treatment |
| **Triglycerides** | ≥1.7 mmol/L | ≥1.7 mmol/L or treatment[b] |
| **High Density Lipoprotein** | ≤1.03 mmol/L (male) or ≤1.29 mmol/L (female) | ≤1.03 mmol/L (male) or ≤1.29 mmol/L (female) |

Notes: ATPIII, National Cholesterol Education Program Third Adult Treatment Panel; IDF, International Diabetes Foundation; WC, waist circumference; BMI, body mass index; SBP, systolic blood pressure; DBP, diastolic blood pressure. ATPIII criteria is met if ≥3 of 5 components present; IDF criteria is met with central obesity plus ≥2 of remaining 4 components.

[a] Diagnosis of diabetes, treatment for diabetes or HbA1c ≥39 mmol/mol (5.7%) used as surrogate for raised fasting glucose

[b] Fibrates or Nicotinic Acid

The methodology for lipid and HbA1c analysis has been described in detail previously [18, 19]. In short, biomarker concentrations were measured at the Biochemistry Department of St James's Hospital, Dublin which is fully accredited to ISO 15189:2012 standard with quality of the assays monitored by internal quality controls and participation in External Quality Control Assessment Schemes.

Blood pressure (BP) [20], waist circumference, height and weight were measured, and body mass index (BMI) calculated as previously described [21]. BMI ≥25kg/m$^2$ and <30kg/m$^2$ was considered overweight while BMI ≥30kg/m$^2$ was considered obese.

'IR', a self-reported doctor-diagnosed diagnosis of diabetes, treatment for diabetes as identified by using the WHO Anatomic Therapeutic Classification (ATC) codes (A10A [insulin] and A10B [non-insulin hypoglycaemics]) or HbA1c ≥39 mmol/mol (5.7%), the lower limit for prediabetes as per the American Diabetes Association [22], was used as surrogate for raised fasting glucose (≥5.6 mmol/L).

For the ATPIII criteria those who had ≥3 of the 5 components were deemed to have MetS, while for the IDF criteria those who had central obesity—BMI >30 kg/m$^2$ or a WC of ≥94 cm (male) or ≥80 cm (female)–plus ≥2 of the remaining 4 components were deemed to have MetS (Table 1).

## Covariates

Data relating to participants' age, sex, educational attainment, and level of physical activity were all collected as part of the CAPI. Physical activity was measured using the International Physical Activity Questionnaire short form, a validated tool to quantify physical activity, with participants categorised as having low, moderate or high levels of physical activity [23]. These covariates were used for the logistic regression models.

Data relating to participants' smoking and chronic disease history were self-reported (CAPI). Smoking was categorised based on smoking history. CVD conditions consisted of angina, heart attack, heart failure, stroke, and transient ischaemic attack. Chronic conditions included incontinence, arthritis, asthma, Parkinson's disease, back pain, cancer, cataracts, glaucoma, liver disease, osteoporosis, and peptic ulcer disease. Hypertension or diabetes were not included in either chronic disease variable given they were already considered as part of

the MetS criteria. Antidepressant use was identified by using the WHO ATC codes (N06AB, N06AX16, N06AX21, N06AX11, N06AX22 and N06AX12). Frailty was operationalised using Fried's frailty phenotype [24] using data from both the CAPI and HA, as described previously [25].

Biomarkers included estimated glomerular filtration rate (eGFR), using both creatinine and cystatin C measurements [26], as an estimate of renal function, with the combination of creatinine and cystatin C demonstrating greater precision than equations using either alone, including in older adults [27]; C Reactive Protein (CRP) was used as a measure of inflammation, with CRP concentrations measured on a Roche Cobas c 701 analyser with a proprietary immunoturbidimetric assay (Roche Diagnostics Ireland, Tina-quant® C-Reactive Protein 3rd Gen); Vitamin D concentration and low-density lipoprotein (LDL) levels were measured as described previously [18, 28]. These covariates, along with BMI, were included for the purpose of comparison of characteristics. BMI was used for comparison rather than WC given the differing WC cut-off points by sex and with males over-represented in those with MetS.

## Statistical analysis

Stata/MP 14.1 software was used for all statistical analysis (StataCorp, College station, TX). Population weights were used to adjust for those who did not take part in the HA and compared to the Central Statistics Office census data for 2011 allowing the sample to be as close to population-representative as possible. The prevalence of MetS was determined, overall and by subgroups stratified by age, sex, educational attainment, and physical activity. Between group differences for characteristics of those with and without MetS were analysed using ANOVA with adjusted Wald test, and chi-square tests as appropriate. Weighted logistic regression models were used to estimate odds ratios (OR), with 95% confidence intervals (CI) for the association between MetS and age, sex, educational attainment, and level of physical activity. Goodness-of-fit of the logistic regression models was examined using Pearson's chi-square test. A P-value <0.05 was considered statistically significant.

## Ethics

Ethical approval was obtained for each wave from the Faculty of Health Science Research Ethics Committee at Trinity College Dublin. Informed written consent was obtained from all participants.

## Results

Of the 8173 participants aged ≥50 years who completed the CAPI, 5657 (69.2%) completed the HA. In total, 5340 (94.4%) had complete data for all variables of interest in relation to the ATPIII and IDF criteria at wave 1.

In terms of the individual components included within the criteria for MetS, 51% were centrally obese according to ATPIII with nearly 77% centrally obese according to IDF. According to BMI, 42.5% (95% CI: 41.0, 44.0) were overweight, while a further 34.5% (95% CI: 33.1, 35.9) were obese. The prevalence of insulin resistance was 15%, with more than three quarters hypertensive (76%), while 40% had elevated TG, and 16% had reduced HDL. Central obesity was more prevalent among females, while IR, hypertension and raised TG were all more prevalent among males (Table 2).

32.5% had MetS according to the ATPIII criteria (95% CI: 31.1, 34.0) with 39.3% according to the IDF criteria (95% CI: 37.8, 40.8). MetS was seen to be more prevalent in males than females. There was increasing prevalence of MetS with age, lower educational attainment, and lower physical activity levels (Table 3).

**Table 2. Weighted prevalence of individual components of metabolic syndrome, overall and by sex.**

| N = 5340 | | ATPIII[a] | IDF[b] |
|---|---|---|---|
| **Central Obesity** | Overall | 51.19 (49.65, 52.73) | 76.65 (75.37, 77.87) |
| | Male | 48.13 (46.01, 50.25) | 74.46 (72.58, 76.24) |
| | Female | 54.05 (51.93, 56.17) | 78.69 (76.99, 80.31) |
| **Insulin Resistance** | Overall | 15.05 (13.93, 16.24) | |
| | Male | 18.09 (16.39, 19.93) | |
| | Female | 12.21 (10.85, 13.72) | |
| **Hypertension** | Overall | 76.54 (75.27, 77.75) | |
| | Male | 82.86 (81.19, 84.41) | |
| | Female | 70.63 (68.71, 72.48) | |
| **Raised TG** | Overall | 40.24 (38.68, 41.82) | 40.32 (38.76, 41.90) |
| | Male | 45.90 (43.65, 48.17) | 45.98 (43.73, 48.24) |
| | Female | 34.94 (32.89, 37.05) | 35.03 (32.98, 37.14) |
| **Reduced HDL** | Overall | 15.66 (14.46, 16.94) | |
| | Male | 14.57 (13.10, 16.18) | |
| | Female | 16.68 (15.04, 18.46) | |

Note: Data presented as weighted proportions with percentages with 95% confidence intervals in brackets. Metabolic syndrome (MetS) as per International Diabetes Foundation (IDF) and National Cholesterol Education Program Third Adult Treatment Panel (ATPIII) criteria

[a] Central obesity: Waist circumference of >102 cm (male) or >88 cm (female); Raised TG: Triglycerides ≥1.7 mmol/L

[b] Central obesity: Body mass index >30 kg/m$^2$ or waist circumference of ≥94 cm (male) or ≥80 cm (female); Raised TG: Triglycerides ≥1.7 mmol/L or treatment (Fibrates or Nicotinic Acid and derivatives)

Those with MetS had higher smoking histories, were frailer, with more comorbidities including higher usage of anti-depressants. Those with MetS also had worse renal function, higher levels of inflammation, lower levels of vitamin D, and while they had lower LDL levels, those with MetS had higher ratios of LDL to HDL (Table 4).

**Table 3. Weighted prevalence of metabolic syndrome in older adults in Ireland overall and by subgroups of sex, age, educational attainment, and physical activity levels.**

| Category | | N | ATPIII | IDF |
|---|---|---|---|---|
| | Overall | 5340 | 32.53 (31.07, 34.03) | 39.29 (37.81, 40.78) |
| **Sex** | Male | 2486 | 35.70 (33.68, 37.76) | 44.16 (42.11, 46.23) |
| | Female | 2854 | 29.57 (27.54, 31.68) | 34.72 (32.61, 36.90) |
| **Age (Years)** | 50–59 | 2264 | 26.61 (24.58, 28.75) | 33.67 (31.51, 35.91) |
| | 60–69 | 1805 | 34.38 (32.04, 36.79) | 40.85 (38.45, 43.30) |
| | ≥70 | 1271 | 39.42 (36.00, 42.95) | 46.04 (42.74, 49.37) |
| **Education** | Primary/None | 1352 | 42.47 (39.49, 45.51) | 49.29 (46.28, 52.30) |
| | Secondary | 2203 | 29.58 (27.53, 31.70) | 36.34 (34.22, 38.51) |
| | Third/Higher | 1784 | 24.76 (22.60, 27.07) | 31.44 (29.15, 33.83) |
| **Physical Activity** | Low | 1552 | 39.25 (36.45, 42.12) | 45.95 (43.17, 48.75) |
| | Moderate | 1874 | 31.95 (29.53, 34.46) | 38.08 (35.63, 40.58) |
| | High | 1871 | 26.64 (24.44, 28.97) | 34.21 (31.87, 36.63) |

Note: Data presented as weighted proportions with percentages with 95% confidence intervals in brackets. Metabolic syndrome (MetS) as per International Diabetes Foundation (IDF) and National Cholesterol Education Program Third Adult Treatment Panel (ATPIII) criteria

**Table 4.  Weighted characteristics of those with metabolic syndrome (MetS) compared to those without MetS.**

| Characteristics | MetS Status | | | MetS Status | | |
|---|---|---|---|---|---|---|
| | ATPIII (n = 1647) | No ATPIII (n = 3693) | p-value | IDF (n = 2001) | No IDF (n = 3339) | p-value |
| Age (Years) | 65.5 (64.7, 66.2) | 62.8 (62.4, 63.3) | p<0.001 | 65.1 (64.4, 65.7) | 62.8 (62.3, 63.3) | p<0.001 |
| Sex (Male, %) | 53.0 (50.5, 55.5) | 46.0 (44.4, 47.7) | p<0.001 | 54.3 (52.0, 56.6) | 44.4 (42.7, 46.2) | p<0.001 |
| Education (%) | | | | | | |
| Primary | 40.7 (37.7, 43.6) | 26.5 (24.6, 28.5) | p<0.001 | 39.1 (36.4, 41.7) | 26.0 (23.9, 28.1) | p<0.001 |
| Secondary | 42.1 (39.3, 44.9) | 48.3 (46.4, 50.2) | | 42.8 (40.3, 45.4) | 48.5 (46.5, 50.5) | |
| Third + | 17.2 (15.4, 19.2) | 25.2 (23.6, 26.9) | | 18.1 (16.4, 20.0) | 25.5 (23.9, 27.3) | |
| Physical Activity (%) | | | | | | |
| Low | 38.1 (35.2, 41.1) | 28.3 (26.5, 30.3) | p<0.001 | 36.9 (34.4, 39.5) | 28.0 (26.1, 30.1) | p<0.001 |
| Moderate | 34.4 (31.7, 37.1) | 35.2 (33.4, 37.1) | | 33.9 (31.6, 36.3) | 35.6 (33.7, 37.6) | |
| High | 27.5 (24.8, 30.4) | 36.5 (34.3, 38.6) | | 29.2 (26.7, 31.9) | 36.4 (34.2, 38.6) | |
| Body Mass Index (kg/m$^2$) | 32.0 (31.7, 32.3) | 27.0 (26.9, 27.2) | p<0.001 | 31.2 (30.9, 31.4) | 27.0 (26.8, 27.2) | p<0.001 |
| SBP (mmHg) | 140.9 (139.9, 142.0) | 134.6 (133.8, 135.4) | p<0.001 | 141.1 (140.2, 142.1) | 133.7 (132.9, 134.6) | p<0.001 |
| DBP (mmHg) | 84.3 (83.7, 85.0) | 81.6 (81.2, 82.0) | p<0.001 | 84.5 (84.0, 85.1) | 81.2 (80.7, 81.6) | p<0.001 |
| Smoking History (%) | | | | | | |
| Non-smoker | 39.1 (36.4, 41.8) | 44.5 (42.6, 46.4) | p<0.001 | 39.5 (37.0, 42.0) | 44.8 (42.9, 46.8) | p<0.001 |
| Light Ex-smoker | 10.7 (9.3, 12.5) | 16.0 (14.7, 17.4) | | 10.7 (9.4, 12.3) | 16.6 (15.2, 18.1) | |
| Heavy Ex-smoker | 30.6 (28.1, 33.2) | 20.5 (19.2, 22.0) | | 30.5 (28.2, 32.8) | 19.5 (18.1, 21.0) | |
| Current Smoker | 19.6 (17.3, 22.1) | 19.0 (17.4, 20.7) | | 19.3 (17.2, 21.6) | 19.1 (17.5, 20.9) | |
| Frailty Phenotype (%) | | | | | | |
| Non-frail | 55.7 (52.9, 58.5) | 67.9 (66.0, 69.7) | p<0.001 | 57.6 (55.0, 60.1) | 68.1 (66.1, 70.0) | p<0.001 |
| Pre-frail | 39.2 (36.5, 42.0) | 29.3 (27.6, 31.1) | | 37.6 (35.2, 40.2) | 29.2 (27.4, 31.1) | |
| Frail | 5.1 (3.9, 6.6) | 2.8 (2.2, 3.7) | | 4.8 (3.7, 6.2) | 2.7 (2.1, 3.6) | |
| CVD Conditions (%) | | | | | | |
| 0 | 83.4 (81.3, 85.3) | 91.2 (90.1, 92.3) | p<0.001 | 84.5 (82.6, 86.3) | 91.4(90.2, 92.5) | p<0.001 |
| 1 | 11.6 (10.0, 13.5) | 6.7 (5.8, 7.8) | | 10.9 (9.4, 12.5) | 6.7(5.7, 7.8) | |
| ≥2 | 5.0 (3.8, 6.4) | 2.1 (1.5, 2.7) | | 4.6 (3.6, 5.9) | 1.9(1.5, 2.6) | |
| Chronic Conditions | | | | | | |
| 0 | 36.5 (34.0, 39.1) | 43.0 (41.2, 44.8) | p<0.001 | 37.5 (35.1, 39.9) | 43.1 (41.2, 45.0) | p = 0.001 |
| 1 | 34.6 (32.0, 37.2) | 31.2 (29.7, 32.8) | | 34.4 (32.1, 38) | 31.0 (29.3, 32.7) | |
| 2 | 17.5 (15.5, 19.7) | 16.8 (15.4, 18.2) | | 17.2 (15.4, 19.3) | 16.9 (15.4, 18.4) | |
| ≥3 | 11.4 (9.8, 13.2) | 9.0 (8.0, 10.1) | | 10.9 (9.5, 12.6) | 9.0 (8.0, 10.2) | |
| Taking Anti-depressant | 10.7 (9.2, 12.4) | 5.5 (4.7, 6.4) | p<0.001 | 9.5 (8.2, 11.0) | 5.6 (4.8, 6.6) | p<0.001 |
| Biomarker | | | | | | |
| HbA1c (mmol/mol) | 36.7 (36.3, 37.1) | 32.1 (32.0, 32.3) | p<0.001 | 36.2 (35.8, 36.5) | 32.0 (31.9, 32.1) | p<0.001 |
| TG (mmol/L) | 2.4 (2.3, 2.5) | 1.4 (1.4, 1.5) | p<0.001 | 2.4 (2.3, 2.4) | 1.3 (1.3, 1.4) | p<0.001 |
| HDL (mmol/L) | 1.3 (1.3, 1.3) | 1.6 (1.6, 1.7) | p<0.001 | 1.3 (1.3, 1.3) | 1.7 (1.6, 1.7) | p<0.001 |
| eGFR (mL/min/1.73m$^2$) | 72.8 (71.6, 74.1) | 81.1 (80.4, 81.8) | p<0.001 | 74.0 (72.9, 75.1) | 81.3 (80.5, 82.0) | p<0.001 |
| CRP (mg/L) | 4.2 (3.9, 4.6) | 3.1 (2.7, 3.5) | p<0.001 | 4.0 (3.7, 4.3) | 3.2 (2.7, 3.6) | p<0.001 |
| LDL (mmol/L) | 2.7 (2.6, 2.8) | 3.0 (2.9, 3.0) | p<0.001 | 2.7 (2.7, 2.8) | 3.0 (3.0, 3.0) | p<0.001 |
| LDL: HDL | 1.9 (1.9, 2.0) | 2.2 (2.1, 2.2) | p<0.001 | 1.9 (1.9, 1.9) | 2.2 (2.1, 2.2) | p<0.001 |
| HDL: TG | 0.7 (0.6, 0.7) | 1.5 (1.5, 1.6) | p<0.001 | 0.7 (0.7, 0.7) | 1.6 (1.6, 1.6) | p<0.001 |
| Vitamin D (nmol/L) | 50.0 (48.8, 51.2) | 58.2 (57.1, 59.2) | p<0.001 | 50.9 (49.8, 52.1) | 58.5 (57.4, 59.6) | p<0.001 |

Note: SBP, systolic blood pressure; DBP, diastolic blood pressure; CVD, cardiovascular disease; HbA1c, glycated haemoglobin; TG, triglycerides; HDL, high density lipoprotein; eGFR, estimated glomerular filtration rate; CRP, C Reactive Protein; LDL, low density lipoprotein. Data presented as weighted means or weighted proportions with percentages with 95% confidence intervals in brackets. Metabolic syndrome (MetS) as per International Diabetes Foundation (IDF) and National Cholesterol Education Program Third Adult Treatment Panel (ATPIII) criteria. Between group differences were analysed using ANOVA with adjusted Wald test given weighted data, and Chi-Square tests as appropriate.

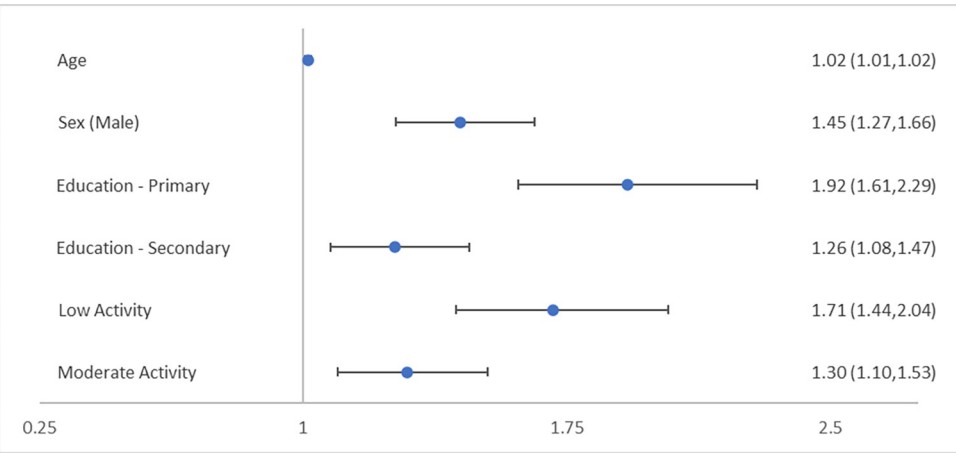

**Fig 2. Categorisation of wave 1 TILDA participants by ATPIII metabolic syndrome (MetS) criteria to estimate odds ratio (95% confidence intervals) for likelihood of MetS.** Third level and higher the base reference for education and high levels of physical activity the base reference for physical activity.

In a weighted analysis of those with IR, 83.7% and 89.2% had MetS by ATPIII and IDF criteria respectively, while of those with reduced HDL, 82.5% and 86.5% had MetS by ATPIII and IDF criteria respectively. Of those who had both IR and reduced HDL, 99.4% (ATPIII) and 98.1% (IDF) were categorised having MetS. Hypertension and central obesity were the individual components with the highest prevalence. Of those who had both hypertension and central obesity as defined by the ATPIII cut-off points, 65.6% (ATPIII) and 65.7% (IDF) had MetS.

Weighted logistic regression models showed that age, sex, educational attainment, and level of physical activity all significantly affected the likelihood of meeting the criteria for MetS. The results of the logistic regression models are summarised in Figs 2 and 3.

Regarding MetS progression with age, with each advancing year the likelihood of MetS increased by 1.5% (0.7–2.3, p<0.001 [ATPIII]) and 1.3% (0.6–2.0, p<0.001 [IDF]).

MetS was more prevalent among males with female sex associated with a 31.0% (21.0–39.7, p<0.001 [ATPIII]) and 38.5% (29.9–46.0, p<0.001 [IDF]) less likelihood of MetS.

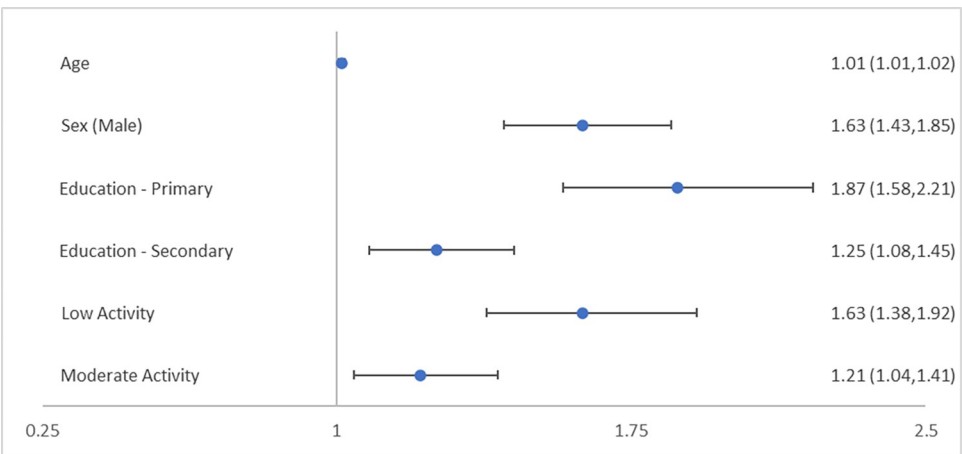

**Fig 3. Categorisation of wave 1 TILDA participants by IDF metabolic syndrome (MetS) criteria to estimate odds ratio (95% confidence intervals) for likelihood of MetS.** Third level and higher the base reference for education and high levels of physical activity the base reference for physical activity.

The levels of education had a significant association on MetS. When compared to having attained a primary level education, a secondary level education reduced the likelihood of MetS by 34.4% (23.1–44.1, p<0.001 [ATPIII]) and 32.9% (21.5–42.6, p<0.001 [IDF]). Additionally, third level or higher educational attainment reduced the likelihood of MetS by 48.0% (38.0–56.4, p<0.001 [ATPIII]) and 46.5% (36.6–54.8, p<0.001 [IDF]).

Low levels of physical activity increased the likelihood of MetS by 71.1% (43.6–103.8, p<0.001 [ATPIII]) and 62.7% (38.1–91.6, p<0.001 [IDF]), when compared to high levels of physical activity. Moderate levels increased the likelihood of MetS by 29.6% (10.0–52.7, p = 0.002 [ATPIII]) and 21.2% (4.3–40.9, p = 0.012 [IDF]), when compared to high levels of physical activity.

Of the 5340 who had all data relevant to MetS at wave 1, 3609 (67.6%) had complete data for all variables of interest in relation to MetS at wave 3. Of those 3609 participants, 1187 (32.9%) met the ATPIII criteria at wave 3 with 991 (27.5%) having done so at wave 1, an increase of 19.8%. 1403 (38.9%) met the IDF criteria at wave 3 with 1223 (33.9%) having done so at wave 1, an increase of 14.7%.

Of the 991 with MetS (ATPIII) at wave 1, 758 (76.5%) had MetS (ATPIII) at wave 3. Of the 2618 without MetS (ATPIII) at wave 1, 429 (16.4%) had MetS at wave 3.

Of the 1223 with MetS (IDF) at wave 1, 933 (76.3%) had MetS (IDF) at wave 3. Of the 2386 without MetS (IDF) at wave 1, 470 (19.7%) had MetS at wave 3.

## Discussion

In this, the first large population-representative study to report the prevalence of MetS in older adults in Ireland, we observed that nearly 2 in every 5 (IDF) and nearly 1 in 3 (ATPIII) people meet the criteria for MetS. When weighted the sample was representative of over 1.19 million community-dwelling adults aged ≥50 years in Ireland, which equates to approximately 480,000 people meeting the ATPIII or IDF criteria for MetS, and is considerably higher than previous Irish estimates [11]. This study demonstrated a 7% higher prevalence of MetS when IDF criteria, rather than ATPIII criteria, are applied, a difference that has been found in similar studies [11, 29]. The high prevalence of MetS in ageing individuals has serious potential implications for both the health of the population and the utilisation of healthcare resources into the future given both the growth of ageing populations and that MetS is a condition that increases the risk of CVD, diabetes, and all-cause mortality.

In terms of individual components of MetS the high prevalence of central obesity was particularly note-worthy. The Survey of Health, Ageing and Retirement in Europe (SHARE), using data from more than 35,000 adults aged ≥50 years across 10 European nations in 2011 reported 60.5% had a BMI ≥25kg/m$^2$ [30]. This is markedly lower than the prevalence of overweight/obese measured in our study where 77% had a BMI ≥25kg/m$^2$. The prevalence of obesity in Ireland, according to BMI (BMI ≥30kg/m$^2$), was 34.5% compared to 19.2% (SHARE) (S1 Table). Results from SHARE's 2011 wave was chosen here as it is a comparable time point to wave 1 of TILDA (October 2009 to February 2011), from which the data from our study was taken, however, the results from a total of four previous and subsequent waves of SHARE (2005, 2007, 2011 and 2013) show overall prevalence of overweight/obese to be reasonably consistent, ranging from 60.1% to 60.5%. The country with the highest point prevalence was Spain (2007) at 73.6%, which is still lower than that observed in our study among older Irish adults. SHARE used self-reported height and weight to calculate BMI which may explain some of the difference, given that weight is under-estimated, and height over-estimated when self-reported, leading to under-estimations of prevalence once BMI is calculated [31]. For further comparison, the prevalence of those with BMI ≥25kg/m$^2$ was 71.6% among adults aged ≥60

years in the United States (2011–2012) using measured height and weight as part of the National Health and Nutrition Examination Survey (NHANES). 35.4% had a BMI $\geq$30kg/m$^2$ [32]. Overall, these results suggest that the prevalence of overweight/obese in older adults in Ireland is high by international standards.

Examination of (ATPIII) MetS prevalence in the United States using data from NHANES showed a prevalence of 46.7% among those $\geq$60 years [33], which is higher than 36.7% for those aged $\geq$60 years in this study. Their results also differed in that there was significantly higher prevalence among women with the largest difference between sexes being among those aged $\geq$60 years. Our study has shown MetS to be more prevalent among males, despite central obesity being more prevalent among females. The differences between these studies may be explained by differences in race/ethnicity between NHANES and TILDA datasets–NHANES noted differences between ethnicities but did not stratify these subgroups by sex. Previous studies have shown that there are differences in prevalence observed between sexes depending on race, with MetS being more prevalent in African-American, Hispanic-American and Indian women than in their male counterparts, by 57%, 26% and 35% respectively [34, 35], but more prevalent among northern European men than their female counterparts [36, 37]. This suggests that genetics and sex hormones may influence MetS prevalence if not due to ethnically driven cultural behaviours.

Chronic subclinical inflammation is known to be a component of MetS [38]. Pro-inflammatory and pro-thrombotic biomarkers such as CRP, IL-6, TNF-$\alpha$, fibrinogen and plasminogen activator inhibitor-1 have all been found to be associated with MetS, however these relationships and potential role in pathogenesis are not well understood [39, 40]. Subcutaneous adipose tissue biopsies from subjects with MetS have been shown to have a 2.5-fold increase in mast cells when compared to controls. Mast cells were positively corelated with components of MetS, such as WC, raised TG and insulin resistance, as well as inflammatory markers such as IL-1$\beta$ and IL-6, suggestive of an inflammatory role in pathogenesis of MetS [41]. In this study those with MetS had higher levels of CRP, a surrogate of inflammatory status, albeit with weighted means that would be considered within clinically 'normal ranges' ($<$5 mg/L), limiting its use on a practical basis. It would be informative to examine levels of other inflammatory biomarkers for MetS as well as ageing, to further investigate the involvement of inflammation in the development of MetS, and associated comorbidities, in aged populations.

Those with MetS had lower LDL levels, which would appear paradoxical given the known associations with risk of CVD for both MetS and LDL. LDL has long been felt to be the predominant atherogenic lipoprotein [42]. LDL levels have previously been shown to be normal in those with MetS but with LDL particles that are smaller and denser than usual [43].

Renal function was lower in those with MetS, likely to be explained at least in part by those with MetS being older and renal function known to decline with age. Vitamin D concentrations were lower in those with MetS, consistent with previous studies [44], and which may be explained by vitamin D being fat-soluble and those with MetS having higher BMI, with more of their vitamin D being sequestered in adipose tissue [45, 46].

Nearly 20% of those with MetS were current smokers. A further 30% were deemed to be 'heavy ex-smokers', while those with MetS were also less likely to have been non-smokers. This is clinically significant in that smoking will have an additive effect to their cardiovascular risk profile.

With regards to the longitudinal findings, nearly 25% of those who had MetS at wave 1 did not have MetS at wave 3. This shows that a diagnosis of MetS and its associated risk profile is reversible. In saying that it is also worth noting that more participants had MetS at wave 3 than at wave 1, a finding that could at least be partially explained by the group ageing by 4 years between waves. This net increase was despite participants being informed at wave 1 of the

results of their height and weight measurements, along with their BMI and what category (underweight, normal, overweight, obese) that represented. Therefore, despite participants being made aware of their baseline BMI, among those who completed the HA at both wave 1 and 3, an increased number met the criteria for MetS at 4-year follow-up despite having the opportunity to make positive 'healthy' lifestyle changes (e.g, increased physical activity or dietary modifications) in the intervening period.

In an ageing society it should be noted that diabetes, (midlife) hypertension and (midlife) obesity, all diagnostic components of MetS, along with smoking, physical inactivity, lower educational attainment, and depression, all of which have been shown to be associated with MetS in this study, have been attributed to approximately one third of Alzheimer's disease (AD) cases [47]. Studies have shown that age-adjusted incidence of dementia has declined or stabilised in recent years, with improved management of hypertension and diabetes being suggested as contributors to this [48]. It has been suggested that delaying the onset of AD by a few years could significantly reduce its prevalence and the associated health and economic burden [49]. A randomised control trial using a multidomain lifestyle intervention including physical activity, dietary counselling and metabolic risk monitoring has shown beneficial cognitive effects in at-risk older participants [50]. Clinicians need to be aware of the scale of the problem in the first instance and be cognisant to identify those at-risk people so that interventions can begin.

In terms of a potential simple clinical screening tool, using the individual components with the highest prevalence, hypertension (76.5%) and central obesity (51.2% using the ATPIII cut-off points), those who had both hypertension and central obesity had a 66% likelihood of having MetS by both ATPIII and IDF criteria and could prompt measurement of lipids and blood sugars. If screening bloods were undertaken, as mentioned previously, those with either IR or reduced HDL had >80% likelihood of having MetS, with a near 100% likelihood if they had both.

The main strength of this study is that it is based on a large sample designed to be nationally representative, allowing findings to be generalised to community-dwelling adults aged ≥50 years in Ireland, with structured collection of data on a wide range of covariates including medications and lifestyle and highly standardised protocols for the CAPI, HA and laboratory methods. In saying that, the household response at wave 1 was 62%, with 69% of those taking part in HA. While statistical weights have been used to account for this to allow the results to be as representative as possible, they are not a perfect substitute for a 100% response rate with full participation in all aspects of the study.

With regards to the limitations of this study, the single biggest limitation is that IR using HbA1c ≥39 mmol/mol or a diagnosis/treatment of diabetes was used as a surrogate for raised fasting glucose, so the criteria for both IDF and ATPIII are not rigidly adhered to. Conversely, using HbA1c may select those with IR by way of impaired glucose tolerance, who may not be selected by raised fasting glucose.

Another limitation is that hypertension may have been misclassified in different ways; e.g. normotensive participants whose SBP/DBP tested high during the HA, often termed 'white coat hypertension' [51]. From a medication point of view, any participant taking a prescribed anti-hypertensive was classified as hypertensive when in clinical practice some anti-hypertensives have other indications and may be being used for other diagnoses such as heart failure in the absence of pre-existing hypertension or for secondary prevention in diabetes.

MetS was most prevalent among males, older participants, those with least formal education and those with lowest levels of physical activity. The odds of having MetS were observed to increase by more than 1% per year of age, while males were 45% (ATPIII) or 63% (IDF) more likely than females to meet the criteria for MetS. While age and sex are non-modifiable risk

factors, the same cannot be said of physical activity, with high levels of physical activity being associated with less likelihood of MetS. Given that this is a cross-sectional study there is no temporality so it cannot be concluded that MetS is consequent to low levels of physical activity and the participants studied may have low levels of physical activity due to MetS. However, the health benefits of physical activity are well documented, including all diagnostic components for MetS [52], so this is an avenue of research that could be considered as it may potentially lead to an intervention to reduce MetS.

## Conclusion

In this study we report that MetS is highly prevalent in older adults in Ireland with 40% of the 1.2 million community-dwelling population aged $\geq$50 years meeting either the ATPIII or IDF criteria. There was an increased likelihood of meeting the criteria for MetS with increasing age, male sex, lower educational attainment, and lower physical activity.

## Supporting information

**S1 Table. Comparison of prevalence of overweight/obese as measured by body mass index.** Notes: Data presented as weighted proportions with percentages with 95% confidence intervals in brackets. TILDA, The Irish Longitudinal Study on Ageing; SHARE, Survey of Health, Ageing and Retirement in Europe; Body mass index (BMI) measured by self-report in SHARE; Overweight = BMI $\geq$25kg/m$^2$ & <30kg/m$^2$; Obese = BMI$\geq$30kg/m$^2$; Overweight/obese = $\geq$25kg/m$^2$. [a] BMI calculated using measured height and weight; [b] BMI calculated from self-reported height and weight.
(DOCX)

## Author Contributions

**Conceptualization:** Kevin McCarthy, Rose Anne Kenny.

**Data curation:** Kevin McCarthy, Eamon Laird, Aisling M. O'Halloran, Deirdre O'Connor, Román Romero Ortuño.

**Formal analysis:** Kevin McCarthy.

**Funding acquisition:** Rose Anne Kenny.

**Methodology:** Kevin McCarthy, Eamon Laird, Aisling M. O'Halloran, Román Romero Ortuño, Rose Anne Kenny.

**Project administration:** Eamon Laird, Aisling M. O'Halloran, Deirdre O'Connor, Román Romero Ortuño, Rose Anne Kenny.

**Resources:** Eamon Laird, Aisling M. O'Halloran, Rose Anne Kenny.

**Software:** Román Romero Ortuño.

**Supervision:** Aisling M. O'Halloran, Padraic Fallon, Román Romero Ortuño, Rose Anne Kenny.

**Validation:** Kevin McCarthy, Eamon Laird, Aisling M. O'Halloran.

**Writing – original draft:** Kevin McCarthy, Rose Anne Kenny.

**Writing – review & editing:** Kevin McCarthy, Eamon Laird, Aisling M. O'Halloran, Padraic Fallon, Deirdre O'Connor, Román Romero Ortuño, Rose Anne Kenny.

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
