## [Decision Letter · Decision Letter 0]

14 Jul 2022

PONE-D-22-13199An examination of the prevalence of metabolic syndrome in older adults in Ireland: Findings from The Irish Longitudinal Study on Ageing (TILDA)PLOS ONE

Dear Dr. McCarthy,

Thank you for submitting your manuscript to PLOS ONE. After careful consideration, we feel that it has merit but does not fully meet PLOS ONE’s publication criteria as it currently stands. Therefore, we invite you to submit a revised version of the manuscript that addresses the points raised during the review process.

We look forward to receiving your revised manuscript.

Kind regards,

Linglin Xie

Academic Editor

PLOS ONE

Journal Requirements:

Reviewers' comments:

Reviewer's Responses to Questions

**Comments to the Author**

1. Is the manuscript technically sound, and do the data support the conclusions?

Reviewer #1: Yes

Reviewer #2: Yes

2. Has the statistical analysis been performed appropriately and rigorously? 

Reviewer #1: Yes

Reviewer #2: Yes

3. Have the authors made all data underlying the findings in their manuscript fully available?

Reviewer #1: Yes

Reviewer #2: Yes

4. Is the manuscript presented in an intelligible fashion and written in standard English?

Reviewer #1: Yes

Reviewer #2: Yes

5. Review Comments to the Author

Reviewer #1: As the author mentioned, this research includes the first large population-representative study to report the prevalence of metabolic syndrome in older adults in Ireland. It used two representative criteria to calculate prevalence which gives strength in the study. It is a human clinical study with a big cohort using straight research design and showing clear conclusion. It was expected to have in deep discussion with previous reported studies. However, the discussion includes many subjects without depth. About this, there are some review comments which should be added in the discussion.

1. In second paragraph where comparing the current result with 10 European nation, it says the difference found could be because of measuring method. However, the study has been conducted in 2011 and in 10 nation in Europe. It has periodic changes as well as the race difference or dietary difference. The author should find better or more evidence to support his idea.

2. In fourth paragraph starting with “Chronic subclinical..”, the author mentioned about how inflammatory status important factor in MetS. Previous publications on human clinical publications examining the relationshp between inflammatory biomarkers and MetS should be addressed as well as which biomarkers they used.

3. In sixth paragraph starting with “Renal function .. “, the author said “Vit D levels were lower in those with MetS”, “which may be explained by Vit D being fat-soluble and those with MetS having higher BMI”. This is confusing and misleading. Why people with higher BMI which is expected to have higher fat percentage will have lower Vit D because it’s fat-soluble? It could be because MetS has lower outdoor activity or lower consumption of milk.

4. In seventh paragraph starting with “With regards .. “, the author said “nearly 25% of those who had MetS at wave 1 did not have MetS at wave 3” by “being informed”. If the author has extra information on the subjects behavior or diet change after being informed, it will be informative to mentioned in the paragraph. In addition, there’s no data shown on wave 1 or 3. I am wondering which wave the data presented in the manuscript is from.

Reviewer #2: This article reads well. It reported MetS is highly prevalent in adults in the age of over 50 years with 40% of the 1.2 million population. The analysis and discussion of the results are comprehensive and very persuasive. It is profound that, other than the USA, this article showed that Mets is more prevalent among northern European men than their female counterparts, which indicating that genetics, hormones, race etc. influence MetS prevalence. In addition, compared with previous study, this article analyzed the prevalence of MetS among older people, which could serve as a guide to prevent MetS, therefore improve quality of life and increase lifespan during aging more precisely.

6. PLOS authors have the option to publish the peer review history of their article (what does this mean?). If published, this will include your full peer review and any attached files.

Reviewer #1: No

Reviewer #2: No

---

## [Author Response · Author response to Decision Letter 0]

26 Jul 2022

I thank the Reviewers and Editor for the detailed review and have addressed their queries in the responses below. 

With regards to the specific queries raised by reviewer 1:

1. In second paragraph where comparing the current result with 10 European nation, it says the difference found could be because of measuring method. However, the study has been conducted in 2011 and in 10 nation in Europe. It has periodic changes as well as the race difference or dietary difference. The author should find better or more evidence to support his idea.

Response: SHARE is the best available study that has examined BMI levels among older adults (aged ≥50 years) in Europe, and most comparable to TILDA in terms of age and available results. It includes countries from northern, central and southern Europe (Austria, Belgium, Denmark, France, Germany, Italy, Netherlands, Spain, Sweden, and Switzerland) with varying diets and cultures. Results from SHARE’s 2011 wave was chosen for discussion as it is a comparable time point to wave 1 of TILDA (October 2009 to February 2011), from which the data from our study was taken. Of the 10 countries included in SHARE, Spain had the highest prevalence of overweight/obese at 70.5%, lower than the prevalence in Ireland noted in our study (77%). While I accept that prevalence of overweight/obesity may fluctuate periodically, the results from a total of four previous and subsequent waves of SHARE (2005, 2007, 2011 and 2013) show prevalence of overweight/obese to be reasonably consistent, ranging from 60.1% to 60.5% across all 10 nations, with the highest point prevalence in Spain in 2007 at 73.6%, which is still lower than that observed in our study among older Irish adults. Similarly, Spain was the country with the highest point prevalence of obesity, at 26.4% (2007) which is considerably lower than the prevalence of obesity in our study (34.5%).

I have also added a note referring to data from the 2011-2012 National Health and Nutrition Examination Survey (NHANES) in the USA to compare their overweight and obesity prevalence, albeit their cohort is aged ≥60 years rather than ≥50 years (reference 32).

2. In fourth paragraph starting with “Chronic subclinical..”, the author mentioned about how inflammatory status important factor in MetS. Previous publications on human clinical publications examining the relationshp between inflammatory biomarkers and MetS should be addressed as well as which biomarkers they used.

Response: I have elaborated further on this topic and included three additional references to studies reviewing or examining the relationships between MetS and inflammation (references 39, 40 and 41).

3. In sixth paragraph starting with “Renal function .. “, the author said “Vit D levels were lower in those with MetS”, “which may be explained by Vit D being fat-soluble and those with MetS having higher BMI”. This is confusing and misleading. Why people with higher BMI which is expected to have higher fat percentage will have lower Vit D because it’s fat-soluble? It could be because MetS has lower outdoor activity or lower consumption of milk.

Response: There are numerous reasons why one person may have lower vitamin D levels than another – from the use of sunscreen or level of outdoor activity/sun exposure affecting cutaneous synthesis to the use of fortified products or supplementation influencing dietary intake. However, those who are obese, such as many of those with MetS, are known to have lower bioavailable vitamin D from both cutaneous and dietary sources because it is sequestered in adipose tissue. I have elaborated on this slightly in the manuscript and added two references (references 45 and 46), which I hope clarifies the matter.

4. In seventh paragraph starting with “With regards .. “, the author said “nearly 25% of those who had MetS at wave 1 did not have MetS at wave 3” by “being informed”. If the author has extra information on the subjects behaviour or diet change after being informed, it will be informative to mentioned in the paragraph. In addition, there’s no data shown on wave 1 or 3. I am wondering which wave the data presented in the manuscript is from.

Response: This study is largely based on wave 1 of TILDA (October 2009 to February 2011). The data collected as part of that wave was used to characterise and determine the national prevalence of MetS among adults aged ≥50 years. 

We also used data from wave 3 of TILDA (March 2014 to October 2015) to investigate trajectories of those with and without baseline MetS, among those who had all relevant data for MetS at both waves, i.e. what proportion of those with baseline MetS did not meet the criteria at wave 3 and vice versa. 

While an observational study, participants at wave 1 of TILDA were informed of the result of their baseline height and weight measurements along with the BMI that corresponded to and what category (underweight, normal, overweight, obese) that represented. Therefore, despite participants being made aware of their baseline BMI, among those who participated at HA at both waves 1 and 3, an increased number met the criteria for MetS at 4-year follow-up despite having the opportunity to make positive ‘healthy’ lifestyle changes (e.g., increased physical activity or dietary modifications) in the intervening period. These possible behavioural changes are not considered in this study.

---

## [Decision Letter · Decision Letter 1]

19 Aug 2022

An examination of the prevalence of metabolic syndrome in older adults in Ireland: Findings from The Irish Longitudinal Study on Ageing (TILDA)

PONE-D-22-13199R1

Dear Dr. McCarthy,

We’re pleased to inform you that your manuscript has been judged scientifically suitable for publication and will be formally accepted for publication once it meets all outstanding technical requirements.

Kind regards,

Linglin Xie

Academic Editor

PLOS ONE

Additional Editor Comments (optional):

Reviewers' comments:

Reviewer's Responses to Questions

**Comments to the Author**

1. If the authors have adequately addressed your comments raised in a previous round of review and you feel that this manuscript is now acceptable for publication, you may indicate that here to bypass the “Comments to the Author” section, enter your conflict of interest statement in the “Confidential to Editor” section, and submit your "Accept" recommendation.

Reviewer #1: All comments have been addressed

Reviewer #2: All comments have been addressed

2. Is the manuscript technically sound, and do the data support the conclusions?

Reviewer #1: Yes

Reviewer #2: Yes

3. Has the statistical analysis been performed appropriately and rigorously? 

Reviewer #1: I Don't Know

Reviewer #2: I Don't Know

4. Have the authors made all data underlying the findings in their manuscript fully available?

Reviewer #1: Yes

Reviewer #2: Yes

5. Is the manuscript presented in an intelligible fashion and written in standard English?

Reviewer #1: Yes

Reviewer #2: (No Response)

6. Review Comments to the Author

Reviewer #1: Thank you for addressing the questions which are been raised. Please check newly added citations if they are correctly cited.

Reviewer #2: (No Response)

7. PLOS authors have the option to publish the peer review history of their article (what does this mean?). If published, this will include your full peer review and any attached files.

Reviewer #1: No

Reviewer #2: No

---

## [Editor Report · Acceptance letter]

23 Aug 2022

PONE-D-22-13199R1 

An examination of the prevalence of metabolic syndrome in older adults in Ireland: Findings from The Irish Longitudinal Study on Ageing (TILDA) 

Dear Dr. McCarthy:

I'm pleased to inform you that your manuscript has been deemed suitable for publication in PLOS ONE. Congratulations! Your manuscript is now with our production department. 

Kind regards, 

on behalf of

Dr. Linglin Xie 

Academic Editor

PLOS ONE